Original research

# Changes in neonatal admissions, care processes and outcomes in England and Wales during the COVID-19 pandemic: a whole population cohort study

Sam F Greenbury,[1,2] Nicholas Longford,[1] Kayleigh Ougham,[1] Elsa D Angelini,[2] Cheryl Battersby,[1] Sabita Uthaya ![ORCID],[1] Neena Modi ![ORCID] [1]

[1]Section of Neonatal Medicine, School of Public Health, Faculty of Medicine, Imperial College London, London, UK
[2]Institute for Translational Medicine and Therapeutics Data Science Group, Imperial College London, London, UK

**Correspondence to**
Professor Neena Modi;
n.modi@imperial.ac.uk

## ABSTRACT

**Objectives** The COVID-19 pandemic instigated multiple societal and healthcare interventions with potential to affect perinatal practice. We evaluated population-level changes in preterm and full-term admissions to neonatal units, care processes and outcomes.

**Design** Observational cohort study using the UK National Neonatal Research Database.

**Setting** England and Wales.

**Participants** Admissions to National Health Service neonatal units from 2012 to 2020.

**Main outcome measures** Admissions by gestational age, ethnicity and Index of Multiple Deprivation, and key care processes and outcomes.

**Methods** We calculated differences in numbers and rates between April and June 2020 (spring), the first 3 months of national lockdown (COVID-19 period), and December 2019–February 2020 (winter), prior to introduction of mitigation measures, and compared them with the corresponding differences in the previous 7 years. We considered the COVID-19 period highly unusual if the spring–winter difference was smaller or larger than all previous corresponding differences, and calculated the level of confidence in this conclusion.

**Results** Marked fluctuations occurred in all measures over the 8 years with several highly unusual changes during the COVID-19 period. Total admissions fell, having risen over all previous years (COVID-19 difference: −1492; previous 7-year difference range: +100, +1617; p<0.001); full-term black admissions rose (+66; −64, +35; p<0.001) whereas Asian (−137; −14, +101; p<0.001) and white (−319; −235, +643: p<0.001) admissions fell. Transfers to higher and lower designation neonatal units increased (+129; −4, +88; p<0.001) and decreased (−47; −25, +12; p<0.001), respectively. Total preterm admissions decreased (−350; −26, +479; p<0.001). The fall in extremely preterm admissions was most marked in the two lowest socioeconomic quintiles.

**Conclusions** Our findings indicate substantial changes occurred in care pathways and clinical thresholds, with disproportionate effects on black ethnic groups, during the immediate COVID-19 period, and raise the intriguing possibility that non-healthcare interventions may reduce extremely preterm births.

## Strengths and limitations of this study

► Our study is a complete population evaluation that included all admissions to National Health Service neonatal units in England and Wales over an 8-year period.

► We assessed full-term, as well as extremely preterm, very preterm and moderate-to-late preterm groups individually.

► All previous studies have compared a COVID-19 period with earlier periods with the implicit assumption that COVID-19 is the only agent likely to have influenced outcomes; however, we show clearly there have been marked fluctuations in outcomes over time, hence assessed differences between the first national COVID-19 lockdown period and the preceding quarter, and compared these with corresponding differences in the previous 7 years.

► A limitation of our approach is that our measure of exceptionality may be too conservative, potentially hindering detection of a COVID-19 effect.

► We were unable to evaluate national data on births by gestational age directly as these were not available.

## INTRODUCTION

The COVID-19 pandemic, the consequence of the emergence of a novel virus, SARS-CoV-2, has had potential to affect maternal and newborn health in multiple ways. In the UK, the first full national lockdown commenced on 23 March 2020.[1] This included requiring people to stay at home except for essential reasons, closure of public venues and all non-essential businesses, and prohibition of public gatherings. The national lockdown, and other policies implemented in an attempt to mitigate the spread of the virus, led to changes in hospital and general practitioner care, and alterations in environmental and societal factors. Thus, air quality improved in many highly populated urban areas,[2] but reports of mental stress, domestic violence and child abuse increased.[3 4] On 18

February 2020, National Health Service (NHS) England advised the UK public not to contact their general practitioners, or go to hospital accident and emergency departments, but instead to contact the NHS111 online and telephone service for medical advice.[5] Within hospitals, in addition to the direct consequences of infection, the abrupt onset of the pandemic necessitated rapid implementation of changes in healthcare processes based on standard infection-control policies, without specific knowledge of the transmissibility, pathogenicity and epidemiology of the novel virus. The rapidity of spread led to redeployment of healthcare staff and prioritised allocation of resources, such as personal protective equipment, to areas of greatest need.

There have been eight previous reports evaluating preterm births in relation to the onset of the pandemic: seven describe a reduction[5–12] and one no change.[13] The spontaneous onset of preterm labour is associated with a number of factors, including infection, systemic illness, severe stress and physical injury. From an epidemiological perspective, seasonal effects, socioeconomic factors and population characteristics also affect the preterm birth rate.[14] The pandemic might have additionally influenced rates of elective caesarean section, with and without medical indication, which are an iatrogenic cause of late preterm births, and a well-recognised cause of respiratory and other problems that lead to neonatal unit admission.[15] However, the incidence of births by elective caesarean section varies by population demographics, across healthcare systems and with time. Thus, for many reasons, identifying any causal determinants of preterm birth is problematic.

Our aim in this study was to determine if any 'highly unusual' changes in admissions to neonatal units in England and Wales, care processes and outcomes occurred following the start of the first national lockdown. Recognising the marked fluctuations in these measures over time, we determined if changes in the immediate COVID-19 period, namely April–June 2020, when compared with the preceding quarter, December 2019–February 2020, were highly unusual in relation to differences between equivalent periods over the preceding 7 years. We also determined if any highly unusual changes persisted into the period July–September 2020.

## METHODS

The study was undertaken under approval from the Health Research Authority and Health and Care Research Wales, and with the agreement of all NHS neonatal units in England and Wales. Contributing neonatal units and their clinical leads are listed in online supplemental table S1.

### Data sources

#### Neonatal admissions

We examined the entire population of babies admitted to NHS neonatal units in England and Wales over the period December 2012–September 2020. We obtained information on admissions, including the numbers of suspected and proven SARS-CoV-2 cases for mothers and babies, over

the study period, from the National Neonatal Research Database (NNRD). This is a national information asset containing detailed clinical information extracted from the electronic patient records of all admissions to NHS neonatal units.[16] Data are quality assured to a research standard.[17] As the care of preterm and sick neonates outside of NHS neonatal units is exceptionally rare in the UK, the data comprise the complete population of eligible infants. Neonatal care in England and Wales is delivered in a networked operational model, with babies transferred to higher or lower designation neonatal units according to care needs. Data management procedures for the NNRD therefore include linking episodes of care across neonatal units to provide a complete, single record from admission to discharge for each baby. No additional data management procedures were undertaken for this study.

### Total live births and stillbirths

We obtained data on stillbirths and total live births from the UK Office for National Statistics.[18] The UK definition of stillbirth is when a baby is born dead after 24 completed weeks of pregnancy. A live birth is any baby born with signs of life, regardless of gestational age (GA). If the baby dies before 24 completed weeks, it is called a miscarriage.

### Outcomes

We categorised admissions by GA as defined by the WHO (extremely preterm GA1: $<28^{+0}$; very preterm GA2: $28^{+0}$ to $31^{+6}$; moderate-to-late preterm GA3: $32^{+0}$ to $36^{+6}$; and full term GA4: $\geq37^{+0}$ weeks$^{+days}$), ethnicity, using collapsed NHS codes (Asian; black; white; mixed/other),[19] and Index of Multiple Deprivation (IMD) quintile through mapping of the maternal Lower-layer Super Output Area (LSOA).[20] The IMD is the official measure of relative deprivation for small areas in England, formed by combining information from seven weighted domains (income; employment; education, skills and training; health and disability; crime; housing and services; living environment) to produce an overall measure of deprivation. The LSOA defines an area of similar population size, with an average of approximately 1500 residents or 650 households.

In addition to admissions, we evaluated a range of care processes and key neonatal outcomes. These were: postnatal transfers (downward, from a higher to lower designation neonatal unit; horizontal, to an equivalent designation neonatal unit; upward, from a lower to higher designation neonatal unit); mode of delivery (elective caesarean section; emergency caesarean section); all-cause mortality (early neonatal (days 1–7); late neonatal (days 8–28)); intubation at resuscitation, surfactant administration, ligation of patent ductus arteriosus, bronchopulmonary dysplasia (defined as any respiratory support or supplemental oxygen at 36 weeks' postmenstrual age), death from or surgery for necrotising enterocolitis, severe brain injury (defined as any seizures, hypoxic ischaemic encephalopathy, intracranial haemorrhage, white matter injury, stroke, central nervous system

infection or kernicterus), therapeutic hypothermia; and breast feeding at discharge.

## Analyses

We compared admissions, processes and outcomes for the initial COVID-19 period April–June 2020 (spring) with the preceding period December 2019–February 2020 (winter) (ie, spring minus winter difference), and contrasted these differences with the differences for the corresponding pairs of periods in the preceding years from 2013 (ie, seven sets of paired differences). We made an a priori decision to exclude March 2020, as this represented a period of variable response to the pandemic. We also considered whether any changes between winter and spring 2019–2020 were sustained into July–September 2020 (summer). We did not use data prior to 2013 as complete data for England and Wales were not available. We excluded ethnicity from the analysis for Wales, as these data were not available for 2020. We evaluated differences in absolute numbers as well as differences in rates.

We defined the change in each measure during the initial COVID-19 period, April–June 2020 (spring), as 'highly unusual' if the difference with the period December 2019–February 2020 (winter) was smaller or larger than all previous corresponding differences. We adopted an empirical Bayes approach to provide a post hoc measure of confidence, or relative strength in the estimate of the difference in rates.[21] For each measure and GA category, we held out the two 3-month periods for the COVID-19 difference (ie, the spring (April 2020–June 2020) and winter (December 2019–February 2020) periods). We then used the 14 corresponding pre-COVID-19 spring and winter 3-month periods to estimate the seven background spring–winter differences against which to assess the COVID-19 spring–winter difference. For the 14 pre-COVID-19 3-month periods, we identified posterior distributions over the binomial probabilities, approximating them with Gaussian distributions by moment matching and applying shrinkage assuming the individual 3-month rates are drawn from a common distribution. We then drew 10 000 independent samples from the 14 posterior distributions to yield a posterior distribution for each of the seven spring–winter differences. For the seven sets of 10 000 posterior samples, we evaluated the proportion that did not meet our criterion for 'highly unusual'. This provides an estimate of the probability (the p value) that the COVID-19 period was not 'highly unusual'. We used a 0.05 threshold as a measure of the strength of the evidence for this conclusion.

We present results in tables and figures showing the periods December–February, April–June and July–September by year, highlighting any highly unusual changes.

## Patient and public involvement

The NNRD has been developed in collaboration with parents and former patients; it is overseen by a steering board that includes parent representatives. There was no additional patient or public involvement in this specific study.

## RESULTS

There were 729 363 admissions to neonatal units in England and Wales over the period December 2012–September 2020. We identified marked fluctuations in all measures over the 8 years. However, during the COVID-19 period April–June 2020, in comparison with the preceding period December–February, there were several changes that were both highly unusual and met our strength of evidence threshold (table 1). Admissions fell (COVID-19 period difference: total −1492; previous 7-year difference range: +100, +1617; p<0.001; full-term: −1142; +104, +1178; p<0.001; preterm: −350; −26, +477; p<0.001). The absolute number of admissions in all preterm GA categories over April–June 2020 (7882) was also the lowest for any April–June or December–February period over the previous 7 years (range 8505–9184). The fall in GA1 (extremely preterm) and GA2 (very preterm) admissions, the most immature babies, continued into the period July–September 2020, unlike GA3 (moderate-to-late preterm) and GA4 (full-term) which rose again (figure 1).

There were highly unusual spring–winter falls in GA1 (extremely preterm) admissions in IMD quintile 1, and GA2 (very preterm) admissions in IMD quintiles 1 and 2, though only the latter had a p value below 0.05 (−41; −20, +59; p=0.036). There were highly unusual falls in GA4 (full-term) admissions in IMD quintiles 3, 4 and 5, and additionally in GA3 (moderate-to-late preterm) admissions in IMD 5 (figure 2). The fall in GA1 (extremely preterm) admissions continued into the period July–September. Full-term black ethnicity admissions rose (+66; −64, +35; p<0.001) in spring, and then fell in the summer (figure 3), in contrast to spring reductions in total Asian (−137; −14, +101; p<0.001) and total white (−319; −235, +643: p<0.001) groups (table 1). Transfers to higher designation neonatal units increased (+129; −4, +88; p<0.001). Transfers to lower designation neonatal units decreased (−47; −25, +12; p<0.001).

There were other highly unusual changes. There was a decrease in the number of GA2 (very preterm) babies born by elective caesarean section (−27; −17, +34; p=0.035). The number of GA1 (extremely preterm) babies born in a hospital with a level 3 (neonatal intensive care) unit fell (−40; +3, +71; p=0.027). The percentage of GA2 (very preterm) babies having surgery for necrotising enterocolitis fell (−1.1%; −0.9%, +0.1%; p=0.017). Breast feeding at discharge fell in GA3 (moderate-to-late preterm) babies (−202; −91, +170; p=0.031; −1.7%; −1.1%, +1.5%; p=0.047), but rose in GA4 (full-term) babies (+1.4%; −1.2%, +1.0%; p=0.031).

There were also changes that fulfilled our criteria for 'highly unusual' but did not meet our strength of evidence threshold, and where numbers were small or where a similar sized effect had occurred during the preceding 7

**Table 1** Summary of highly unusual changes in admissions to neonatal units in England and Wales during April–June 2020 (spring), the first 3 months of national COVID-19 lockdown

| Highly unusual changes | Gestational age category | Direction of change (Apr–Jun 2020 compared with Dec 2019-Feb 2020) | Absolute magnitude of change (Apr–Jun 2020 compared with Dec 2019–Feb 2020) | Range of change between Apr–Jun and preceding Dec–Feb in the years 2012–2019 | P value |
|---|---|---|---|---|---|
| Total babies admitted (N) | All preterm | Decrease | −350 | −26, +479 | <0.001 |
| | Full-term | Decrease | −1142 | +104, +1178 | <0.001 |
| | All admissions | Decrease | −1492 | +100, +1617 | <0.001 |
| Black ethnicity (N) | Full-term | Increase | +66 | −64, +35 | <0.001 |
| Asian ethnicity (N) | All admissions | Decrease | −137 | −14, +101 | <0.001 |
| White ethnicity (N) | Full-term | Decrease | −218 | −21, +365 | <0.001 |
| | All admissions | Decrease | −319 | −235, +643 | <0.001 |
| Socioeconomic quintile 2 | Very preterm | Decrease | −41 | −20, +59 | 0.036 |
| Socioeconomic quintile 3 | Full-term | Decrease | −148 | +28, +307 | <0.001 |
| Socioeconomic quintile 4 | Full-term | Decrease | −135 | −39, +198 | <0.001 |
| Socioeconomic quintile 5 (least deprived) | Moderate to late preterm | Decrease | −51 | −8, +58 | <0.001 |
| | Full-term | Decrease | −175 | +17, +164 | <0.001 |
| Elective caesarean section (N) | Very preterm | Decrease | −27 | −17, +34 | 0.035 |
| Elective caesarean section (%) | Very preterm | Decrease | −2.3 | −1.3, +2.0 | 0.035 |
| Born in hospital with level three neonatal unit (intensive care) (N) | Extremely preterm | Decrease | −40 | +3, +71 | 0.027 |
| Transfer to higher designation neonatal unit (N) | Moderate-to-late preterm | Increase | +37 | −8, +18 | 0.007 |
| | Full-term | Increase | +69 | +10, +53 | <0.001 |
| | All admissions | Increase | +129 | −4, +88 | <0.001 |
| Transfer to lower designation neonatal unit (N) | Full-term | Decrease | −15 | −8, +3 | 0.004 |
| | All admissions | Decrease | −47 | −25, +12 | <0.001 |
| Necrotising enterocolitis surgery (%) | Very preterm | Decrease | −1.1 | −0.9, +0.1 | 0.017 |
| Breast feeding at discharge (N) | Moderate-to-late preterm | Decrease | −202 | −91, +170 | 0.031 |
| | Full-term | Decrease | −65 | −38, +267 | 0.015 |
| Breast feeding at discharge (%) | Moderate-to-late preterm | Decrease | −1.7 | −1.1, +1.5 | 0.047 |
| | Full-term | Increase | +1.4 | −1.2, +1.0 | 0.031 |

The p value reflects the uncertainty in the comparison of the spring–winter 2019–2020 differences and spring–winter differences in the previous 7 years; the table lists all results for which the p value is less than 0.05.
We considered a change highly unusual if the difference (whether positive or negative) between this period and December 2019–February 2020 (winter) was greater than the corresponding differences for all 7 preceding years, or was in the opposite direction to all previous differences regardless of magnitude.
%, percentage of infants admitted in gestational age category; N, absolute number.

years, casting uncertainty on their relevance. The number of GA4 (full-term) babies born by emergency caesarean section fell (−186; +45, +500); the percentage requiring intubation at resuscitation rose (+0.3%; −0.5%, +0.15%) as did the proportion with severe brain injury (+0.3%; −0.2%, +0.3%). The percentage of GA1 (extremely preterm) babies receiving surfactant (+2.5%; −1.6%, +1.2%) and the number and percentage of GA2 (very preterm) babies receiving surgery for patent ductus arteriosus (N: +2; −5, +1; %: +0.2%; −0.4%, +0.1%) rose. The percentage of GA3 (moderate-to-late preterm) babies developing bronchopulmonary dysplasia fell (+0.6%; −0.7%, +0.1%). We identified no highly unusual changes in antenatal steroid use, horizontal transfers, therapeutic hypothermia or early and late neonatal mortality. All outcomes evaluated are shown in the online supplemental table S2.

We show the number of suspected and confirmed cases of COVID-19 in mothers and babies over the periods December 2019–February 2020, April 2020–June 2020 and July–September 2020 in table 2. Using Office for National Statistics data, we show changes in stillbirths and live births for England and Wales over the study period; these do not suggest a highly unusual change occurred over April–June 2020 (figure 4).

## DISCUSSION

We identified highly unusual changes in key perinatal measures during the immediate period of the first national UK lockdown, although the number of confirmed cases of COVID-19 in babies admitted to neonatal units, and their mothers, was small. Our study included all admissions to NHS neonatal units in England and Wales over

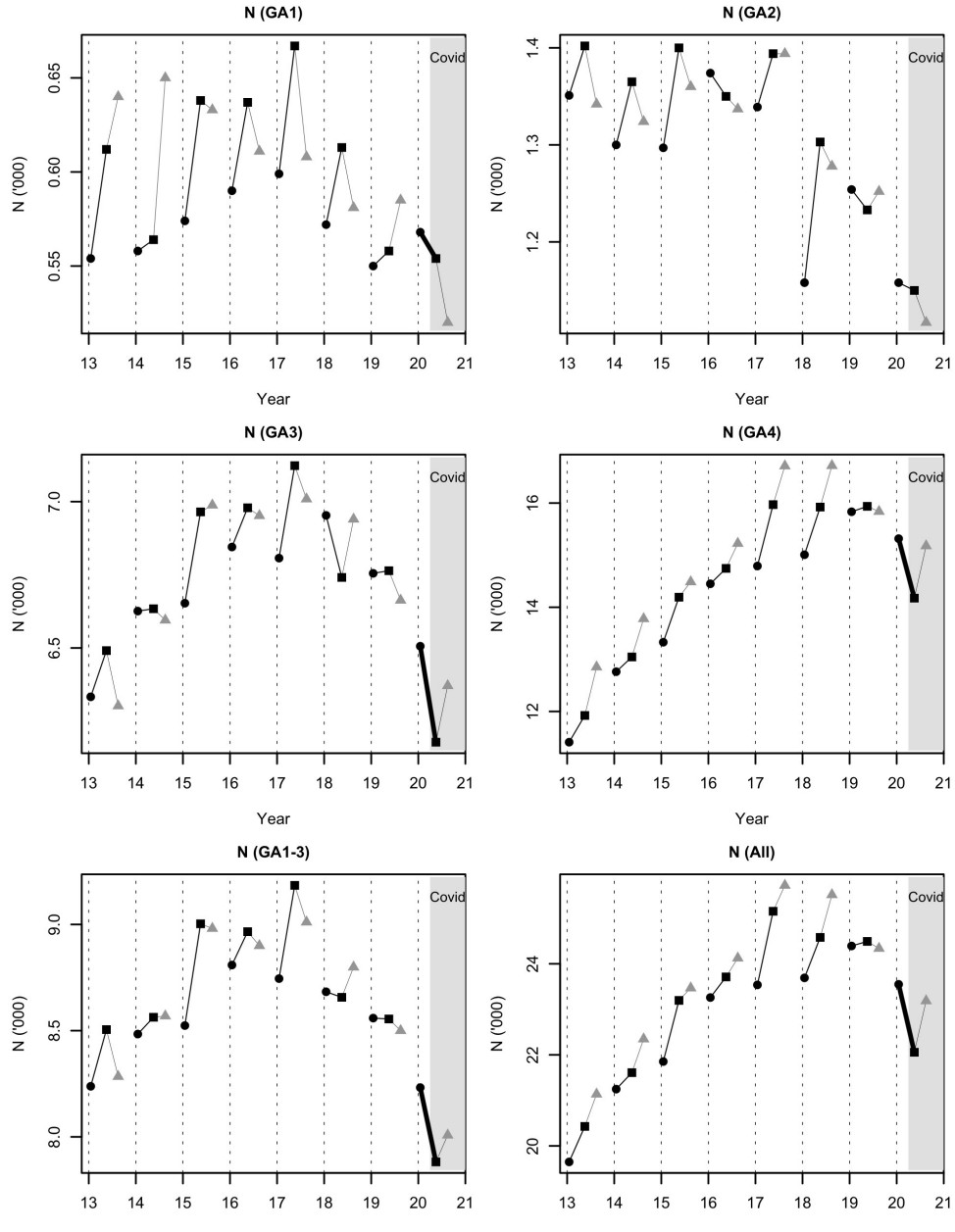

**Figure 1** Admissions to neonatal units in England and Wales by gestational age (GA) category and year. GA1: extremely preterm; GA2: very preterm; GA3: moderate-to-late preterm; GA4: full-term; black circle: December–February; black square: April–June; grey triangle: July–September. The COVID-19 period is highlighted; the thick black lines indicate a change that was highly unusual. There was a highly unusual fall in all preterm (GA groups 1–3 combined) and full-term (GA4) admissions during the period April–June 2020. The falls in GA1 and GA3 admissions were individually also highly unusual; the falls in GA1 and GA2, the most immature babies, continued into the period July–September 2020, unlike GA3 and GA4 which rose again.

an 8-year period. We assessed all preterm and full-term admissions as well as extremely preterm, very preterm and moderate-to-late preterm groups individually, as degree of immaturity has a cardinal influence on care pathways and morbidities. In view of known seasonal fluctuations in births, we assessed the difference between the immediate period of national COVID-19 lockdown with the preceding quarter, excluding a priori the entire month of March 2020, and compared them with differences in the corresponding epochs of the previous 7 years.

We found a highly unusual fall in full-term admissions during the immediate COVID-19 period. This was not due to a fall in total births, or a reduction in elective caesarean sections, following which infants are more likely to require neonatal unit admission than those born vaginally.[15] This suggests a rise in the clinical threshold for the admission of mature babies to neonatal units occurred during the immediate COVID-19 lockdown. Despite the fall in admissions, there was a highly unusual increase in transfers of moderate-to-late preterm and

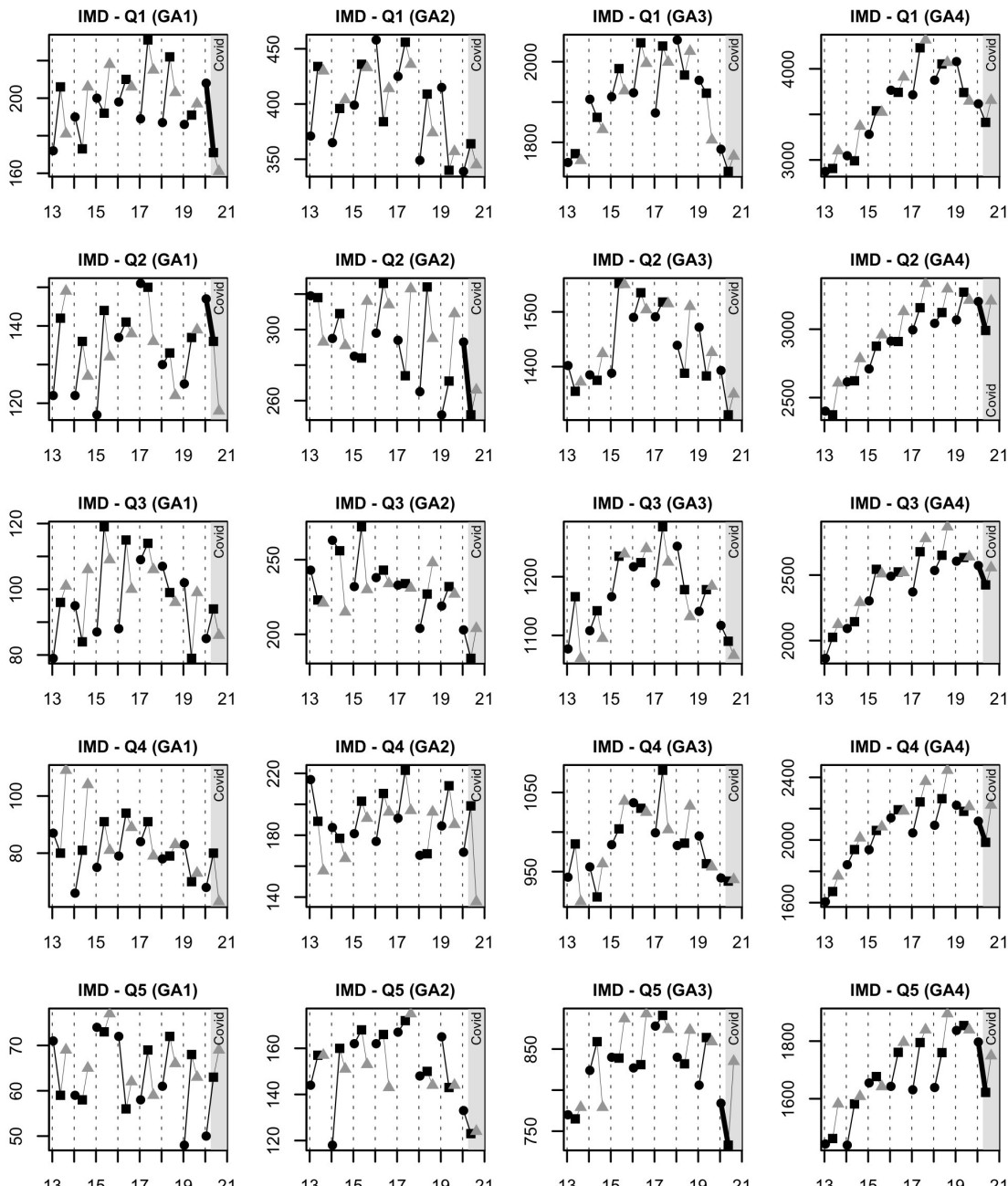

**Figure 2** Admissions to neonatal units in England and Wales by gestational age (GA) category, year and Index of Multiple Deprivation (IMD) quintile. Black circle: December–February; black square: April–June; grey triangle: July–September. The COVID-19 period is shaded; the thick black lines indicate a change that was highly unusual. GA1: extremely preterm; GA2: very preterm; GA3: moderate-to-late preterm; GA4: full-term; Q1: quintile 1 (most deprived); Q5: quintile 5 (least deprived). There were highly unusual falls in GA1 (extremely preterm) admissions in IMD quintiles 1 and 2, and in GA2 (very preterm) admissions in IMD quintile 2 over April–June 2020; the fall in GA1 (extremely preterm) admissions was sustained into the period July–September. In contrast, there was a highly unusual fall in GA3 (moderate-to-late preterm) admissions over the COVID-19 period only in IMD quintile 5 and in GA4 (full-term) admissions in quintiles 2, 3, 4 and 5.

full-term babies to a higher designation neonatal unit. Upward transfer of mature babies is usually only undertaken if higher intensity care is required, suggesting the number with serious illness increased substantially. In this context, the increase in the proportion of full-term babies born by emergency caesarean section, requiring intubation for resuscitation and with severe brain injury should be noted. These changes fulfilled our criteria for highly unusual, although numbers were small and our strength of evidence threshold was not reached. A further notable finding was that the fall in full-term admissions masked a highly unusual increase in the number of admissions of full-term babies of black ethnicity, contrasting with a decrease in Asian and white ethnic groups. Taken together, our data indicate greater likelihood of late presentation and delayed delivery of mature babies in

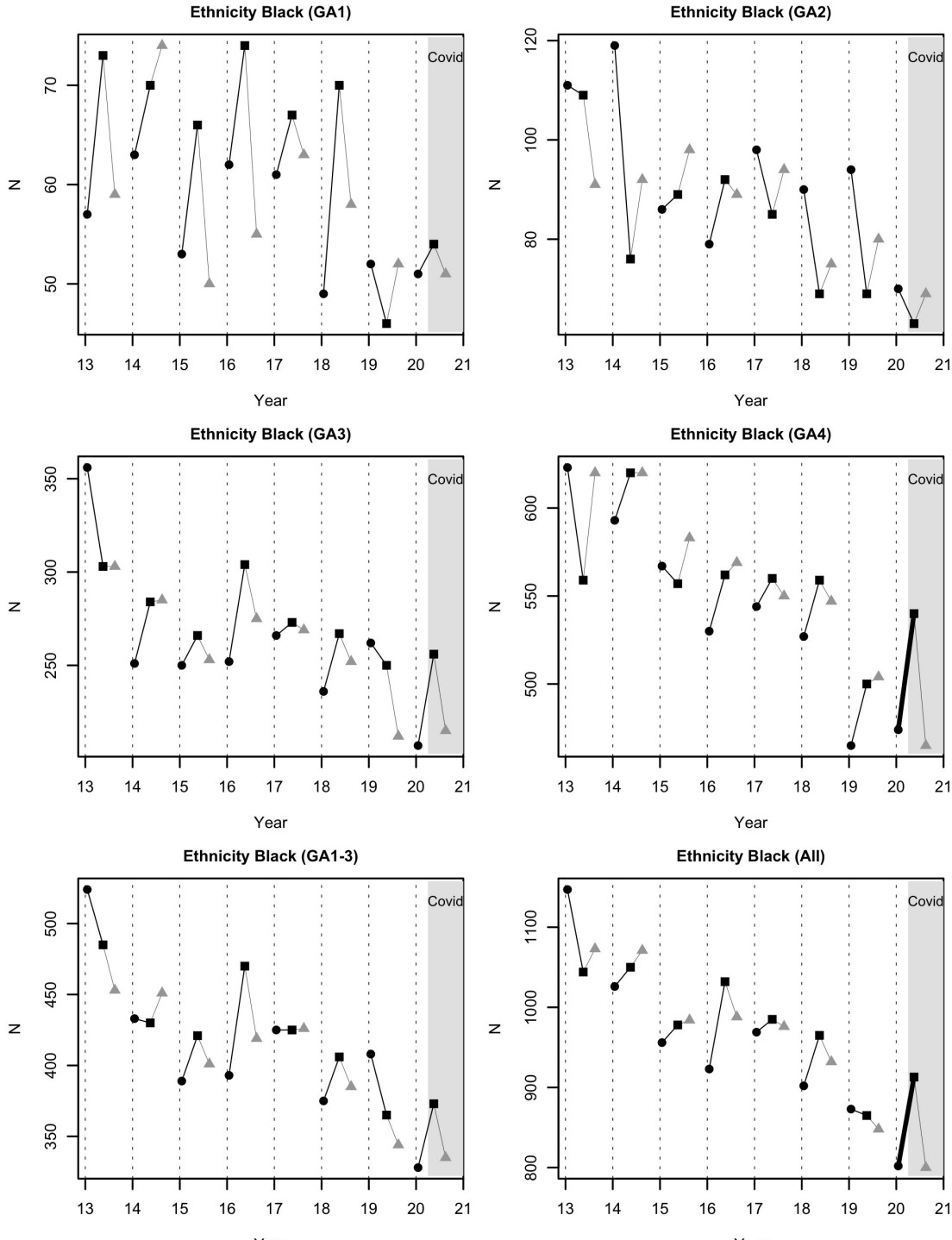

**Figure 3** Admissions of black ethnicity babies to neonatal units in England and Wales by year and period. GA1: extremely preterm; GA2: very preterm; GA3: moderate-to-late preterm; GA4: full-term; black circle: December–February; black square: April–June; grey triangle: July–September. The COVID-19 period is highlighted; the thick black lines indicate a change that was highly unusual. There was a highly unusual increase in all admissions (GA groups 1–4 combined) over April–June 2020, driven by the full-term (GA4) category. This increase was not sustained into the period July–September 2020. GA, gestational age.

fetal distress, in accord with the known marked reduction in all healthcare-seeking behaviours with the onset of the pandemic,[22 23] and greater adverse impact on black communities.[24 25]

We found evidence of other perturbations to neonatal care pathways. It is a UK standard of care to deliver extremely preterm infants in a hospital with a level 3 (neonatal intensive care) neonatal unit.[26] However, during the immediate COVID-19 period, there was a highly unusual decrease in the number of extremely preterm babies born in hospitals with a level 3 neonatal unit. This indicates that obstetric in utero transfers

**Table 2** Numbers of mothers and babies with suspected and confirmed SARS-CoV-2 infection

|  | Mother | | Baby | |
| --- | --- | --- | --- | --- |
|  | Suspected | Confirmed | Suspected | Confirmed |
| Dec 2019–Feb 2020 | 22 | 9 | 46 | 8 |
| Apr 2020–Jun 2020 | 486 | 89 | 139 | 13 |
| Jul 2020–Sep 2020 | 189 | 42 | 20 | 3 |

(transfers of mothers at risk of extremely preterm delivery to a tertiary centre) were less likely. The fall in total admissions meant it was important to evaluate the proportion of babies experiencing a particular outcome. We identified changes that though fulfilling our criteria for highly unusual, and meeting our strength of evidence threshold, were small and may have occurred by chance. These included a decrease in the proportion of very preterm babies receiving surgery for necrotising enterocolitis and an increase in the proportion of full-term babies breast feeding at discharge.

We also identified a highly unusual fall in all preterm admissions, though we were unable to distinguish between spontaneous and medically indicated preterm births. The numbers of moderate-to-late preterm babies dominate the preterm category, and a fall in their admission numbers may, as with full-term babies, reflect a rise in clinical thresholds. However, we also found a highly unusual fall in extremely preterm admissions, those born below 28 weeks' gestation, a change that appeared

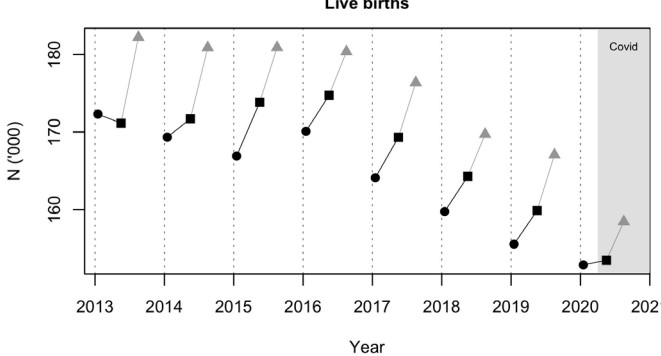

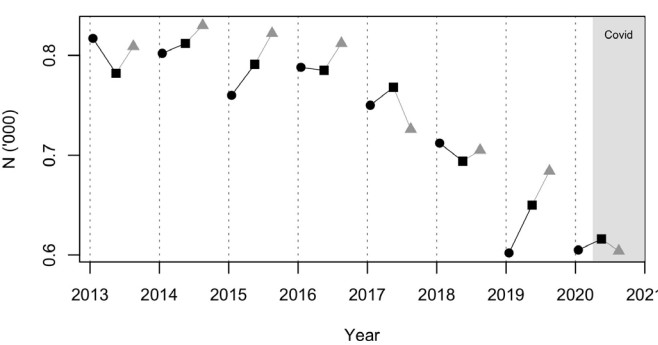

**Figure 4** Live births and stillbirths, England and Wales by 2013–2020 and period. Black circle: December–February; black square: April –June; grey triangle: July–September; The COVID-19 period is highlighted.

confined to the two lowest IMD quintiles representing the most deprived groups. In both, the fall continued into the period July–September 2020. The absolute numbers of extremely preterm babies, even in a whole population dataset, are small, hence it is unsurprising that even though highly unusual, the fall did not meet our stringent statistical threshold. There have however been seven previous reports of a fall in preterm births during the immediate COVID-19 period, though all involved substantially smaller numbers than our study.[6–12] Berghella *et al* compared records from a single hospital in northeast USA over 1 March 1–31 July 2020, with the same period in 2019.[6] They identified 7 births below 28 weeks' gestation in 2020, compared with 14 in the previous year. Philip *et al* compared births at a regional hospital in Ireland over 1 January–30 April 2020 with the same period of the preceding 19 years, identifying only three very and extremely low birthweight infants compared with a predicted number of eight.[7] However, Ireland implemented lockdown measures in early March, not in early January, weakening the inference of a temporal association. Been *et al* used a difference-in-regression-discontinuity approach to study the impact on preterm births of COVID-19 mitigation measures introduced at three points in March 2020 in the Netherlands. They identified a statistically significant reduction only in moderate-to-late preterm births and only in relation to the first time point.[8] Hedermann *et al* compared the period 12 March–14 April 2020 with the average rate in Denmark over the previous 5 years.[9] They identified only 58 extremely preterm births over the 5-year period and noted extremely preterm births were significantly lower in 2020, but not very or moderate-to-late preterm births. They were unable to exclude the possibility of a corresponding rise in late abortions or stillbirths. Matheson *et al* studied births in three maternity hospitals in Melbourne, Australia, identifying 9 extremely preterm births over July–September 2020, compared with 20 during the same period in 2019.[10] Lemon *et al* describe a decrease in preterm births in a single US hospital limited to white women from more advantaged neighbourhoods.[11] Maeda *et al* studied records from 186 Japanese acute care hospitals noting a decrease in preterm births but the 95% CI for the adjusted incidence rate ratios included or were close to one (below 34 weeks' gestation: 0.71; 95% CI: 0.50 to 1.00; below 37 weeks: 0.85; 95% CI: 0.74 to 0.98) and the extent of population coverage is not known.[12] Handley *et al* noted no decrease in preterm births in two

Philadelphia hospitals.[13] However, comparisons between the UK and USA are problematic, first because the healthcare systems are very different, and second, because US reports are centre rather than population based, and hence at risk of ascertainment bias.

All these studies compared a COVID-19 period with earlier periods. In such a direct comparison, there is an implicit assumption that COVID-19 is the only agent likely to have influenced the outcome. However, as we show, there have been marked fluctuations in outcomes over previous years. As the onset and duration of other influences are unknown, subsuming them into the residual error term of a model risks derived a flawed estimate. In contrast to these studies, we considered the *differences* between 3-month pre-COVID-19 and post-COVID-19 periods and compared them with the corresponding 3-month differences over previous 7 years. By comparing differences, we are able to assess the strength of a change during the COVID-19 period taking other, unknown, influences into account. We acknowledge, however, that a limitation of our approach is that our measure of exceptionality may be too conservative, potentially hindering detection of a COVID-19 effect. We also acknowledge that we made no adjustment for multiple comparisons as our p values were used solely for evaluating the relative strength of evidence. Our approach is aligned with other Bayesian approaches[27] and our exploration of population-based data should be regarded more as a hypothesis-generating rather than a hypothesis-testing analysis.

We identified a fall in extremely preterm admissions over April–June 2020 in comparison with December 2019–February 2020, whereas in all previous 7 years the number rose over corresponding periods. In the UK, all extremely preterm babies are admitted to an NHS neonatal unit, hence the fall likely reflects a genuine reduction in live births in this GA group. Though a small study from a single London hospital, employing a before and after approach, suggested stillbirths rose during the immediate COVID-19 period,[28] this is not supported by data from the Office for National Statistics. Our finding that the highly usual reduction in extremely preterm admissions during the immediate COVID-19 national lockdown occurred in the most deprived socioeconomic groups and was sustained into the following 3 months is intriguing. Globally, preterm birth rates are increasing, with a strong association with poverty, disadvantage and deprivation.[29] Attempts to lower the preterm birth rate have remained stubbornly resistant to a range of medical interventions over the years, from widespread use of tocolytics, bedrest, cervical cerclage, vaginal progesterone and enhanced surveillance. Thus, the possibility that non-healthcare-related interventions may be effective is important.

In conclusion, our observation of a fall in extremely preterm admissions during the immediate period of national COVID-19 lockdown, sustained in lower socioeconomic groups into the subsequent 3 months, requires corroboration, and we hope data will be forthcoming from other large, population-based birth cohorts. Our findings should also provide impetus to study the effects on preterm births of public health interventions, such as improved air quality, reduced exposure to crowded environments, altered working during the second trimester of pregnancy, and their interactions with other trigger events, and with socioeconomic status and ethnicity. The reasons for the fall in admissions of more mature babies are more likely to be related to changes in clinical thresholds. Together with evidence of perturbations in care pathways, these findings justify consideration of preparedness and public messaging during national crises adding weight to calls for an official COVID-19 inquiry into UK government actions,[30] such as the recommendation to rely on the call service NHS111 for medical advice,[31] that has now been agreed but deferred until the spring of 2022.[32] Finally, the highly unusual rise in admissions of full-term black ethnicity babies, contrasted with a fall in all other ethnic groups, adds to the growing evidence of a disproportionately higher adverse impact on this demographic group and speaks to the moral imperative to address ethnic and socioeconomic health disparities urgently, as well as growing calls for investment in research to improve maternal and newborn health.[33]

**Acknowledgements** We gratefully acknowledge the contribution of all UK Neonatal Collaborative neonatal units to the National Neonatal Research Database, Mr Richard Colquhoun, research manager, and data analysts Victor Banda and Julia Lanoue.

**Contributors** All authors had full access to all study data and take responsibility for the integrity of the data, the accuracy of the analysis and the decision to submit for publication. The study was conceived by NM, CB and SU. Data were prepared by KO and SFG. The analysis was conducted by NL, EDA and SFG. Figures were prepared by NL. The paper was written by NM. All authors reviewed and contributed to the final draft submitted. The guarantor is NM.

**Funding** We acknowledge funding for SFG through a Medical Research Council award (MR/T016752/1) and for NL through an award from the Health Data Research UK Hub Discover-NOW (reference not applicable), both held by NM.

**Disclaimer** The funders had no role in study design, collection, analysis, and interpretation of data, writing of the report, or the decision to submit the paper for publication.

**Competing interests** None declared.

**Patient consent for publication** Not required.

**Ethics approval** This study was conducted under approval by the UK Research Ethics Service (London Queen Square Research Ethics Committee 21/LO/0024).

**Provenance and peer review** Not commissioned; externally peer reviewed.

**Data availability statement** Data are available upon reasonable request.

**ORCID iDs**
Sabita Uthaya http://orcid.org/0000-0002-6112-2277
Neena Modi http://orcid.org/0000-0002-2093-0681

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
