## [Reviewer comments · BMJ Open]

ARTICLE DETAILS

TITLE (PROVISIONAL)	Changes in neonatal admissions, care processes and outcomes in England and Wales during the COVID-19 pandemic: a whole population cohort study
AUTHORS	Greenbury, Sam; Longford, Nicholas; Ougham, Kayleigh; Angelini, Elsa; Battersby, Cheryl; Uthaya, Sabita; Modi, Neena

VERSION 1 – REVIEW

REVIEWER	Atsushi Miyawaki The University of Tokyo, Department of Public Health, Graduate School of Medicine
REVIEW RETURNED	02-Aug-2021

GENERAL COMMENTS	1. Please add the definition of stillbirths in this study (in the UK). The threshold of stillbirths vs abortions depends on the states (20 or 22 or 24 weeks?). Many countries have reported a decline in the number of pregnancies during the COVID-19 pandemic. If the number of pregnancies that have been established since March has declined, this may explain the downward trend in the absolute number of stillbirths in August and September, depending on the definition of stillbirth. 2. The decrease in very preterm babies in low SES areas is interesting. Based on previous international comparisons, the decrease in preterm births may depend on social context, availability of resources, etc. (https://doi.org/10.1016/S2214-109X(21)00079-6). In addition to neonates, there have been many reports of differences in the effects of COVID-19 by race and SES. In this context, it may be beneficial for readers to provide any potential mechanisms or policy implications for differences by race or SES in the Discussion? 3. I have been more of a frequentist and not that familiar with Bayesian estimation, but the argument based on Bayesian estimation that you are making in this study appears to be correct. This study compares a number of outcomes. Based on my limited knowledge of Bayesian estimation, I don't think it is necessary to consider multiple comparisons in this study (I am not sure if this is correct), but many frequentist readers of the paper may be concerned about this issue (some of the "statistical test" results may be seen as a coincidence).). It may be useful for frequentists to explain this point in a short sentence and citation.
--

REVIEWER	Lara Lemon University of Pittsburgh, Obstetrics, Gynecology and Reproductive Medicine
REVIEW RETURNED	04-Aug-2021

GENERAL COMMENTS	Authors used a clever approach to compare the change in NICU utilization in the quarter pre- and post-COVID to the previous seven-year changes. It is an important paper with strong methods. Detailed comments are below. Major comments: MC1. Limit the number of stratifications. Table 1 is hard to follow and understand the impact with the variety of stratifications. I suggest limiting the gestational age stratifications to: Term, PTB (<37), and VPTB. Using the GA1, etc abbreviations is also confusing to the reader. Or limit the IMD stratifications. -When only the significant changes are displayed it is hard to understand overall effects also. For instance: What happened to breast feeding overall? I would consider recreating Table 1 with all changes assessed, not only those 'unusual', after only a select few stratifications are chosen. Justification for which are chosen should be provided. MC2. Did you run a sensitivity analysis excluding the covid+ deliveries? MC3. I believe your findings of increases in NICU admissions for Black infants is novel and should be highlighted more. Again, (re: MC1) was there an increase in Black babies' admissions regardless of term? I wonder because black women have been seen to have the least impact regarding preterm decreases. MC4. Please describe if you were able to account for spontaneous vs indicated deliveries. If so, this is a very important aspect that should be used in your analysis. Though the 'cause of preterm birth' on page 4 line 40, is largely unknown, discerning between spontaneous and indicated is possible in many data sets. Were authors able to determine this difference? Specifically: SC1. Please explain a bit further if this would include infants admitted directly from delivery? I believe so but am not certain. SC2. Authors have opposite finding from previous studies in US- demonstrating less preterm in the most deprived groups. Is it possible mandates in UK had differential impact compared to those in the US? A sentence on this in the discussion on this is warranted. SC3. Page 3, final bullet point: No gestational age? Or no delivery data? SC4. Would a period effect say from 2013-2015 then have a big impact with this approach to estimate the background? SC5. Methods on page 6 lines 14-30 are confusing. Specifically, how are there 14 pre-COVID differences? I understand 14 time points being both quarters (2013-2020), but only 7 differences, no? Or was each compared to post-COVID? Please elaborate. SC6. To support page 8, lines 8-13: do you have the diagnoses and chief complaint for the admission? SC7. Page 9 lines 57-page 10 lines 5: Please make more clear. I cannot understand this sentence. SC8. Are authors claiming that late care caused an increase in term NICU admissions though overall lower clinical thresholds for admission were a good thing? SC9. Do you have any data on the ed use? This would be an interesting addition. SC10. Did previous period changes include or exclude March? SC11. I suggest changing 'spring-winter' to 'winter-spring' throughout as this is the chronological order for the comparison. Same with table 1. The change is negative because we are going winter to spring, correct?
---

VERSION 1 – AUTHOR RESPONSE

Responses to editor and reviewer comments

Changes in neonatal admissions, care processes and outcomes in England and Wales during the COVID-19 pandemic: bmjopen-2021-054410

Reviewer 1

Dr Atsushi Miyawaki, University of Tokyo

1.1 Please add the definition of stillbirths in this study (in the UK). The threshold of stillbirths vs abortions depends on the states (20 or 22 or 24 weeks?). Many countries have reported a decline in the number of pregnancies during the COVID-19 pandemic. If the number of pregnancies that have been established since March has declined, this may explain the downward trend in the absolute number of stillbirths in August and September, depending on the definition of stillbirth.

Author response: We have added the following to the section “Data sources: Total live and stillbirths” *“The UK definition of stillbirth is when a baby is born dead after 24 completed weeks of pregnancy. A live birth is any baby born with signs of life, regardless of gestational age. If the baby dies before 24 completed weeks, it is called a miscarriage”*.

1.2 The decrease in very preterm babies in low SES areas is interesting. Based on previous international comparisons, the decrease in preterm births may depend on social context, availability of resources, etc. ([https://doi.org/10.1016/S2214-109X\(21\)00079-6](https://doi.org/10.1016/S2214-109X(21)00079-6)). In addition to neonates, there have been many reports of differences in the effects of COVID-19 by race and SES. In this context, it may be beneficial for readers to provide any potential mechanisms or policy implications for differences by race or SES in the Discussion?

Author response: Thank you; we have revised the relevant section in the “Discussion” as follows: *“Our findings should also provide impetus to study the effects on preterm births of public health interventions, such as improved air quality, reduced exposure to crowded environments, altered working during the second trimester of pregnancy, and their interactions with other trigger events, and with socio-economic status and ethnicity”*.

1.3 I have been more of a frequentist and not that familiar with Bayesian estimation, but the argument based on Bayesian estimation that you are making in this study appears to be correct. This study compares a number of outcomes. Based on my limited knowledge of Bayesian estimation, I don't think it is necessary to consider multiple comparisons in this study (I am not sure if this is correct), but many frequentist readers of the paper may be concerned about this issue (some of the "statistical test" results may be seen as a coincidence). It may be useful for frequentists to explain this point in a short sentence and citation.

Author response: The reviewer is correct. We do also acknowledge the debate around the necessity for adjustment for multiple comparisons in both Bayesian and frequentist discussions. We have added the following to the “Discussion” section: *“We also acknowledge that we made no adjustment for multiple comparisons as our p-values were used solely for evaluating the relative strength of evidence. Our approach is aligned with other Bayesian approaches (27) and our exploration of population-based data should be regarded more as a hypothesis generating rather than a hypothesis testing analysis”*

We have also included an additional reference as the reviewer recommends: Sjölander A, Vansteelandt S Frequentist versus Bayesian approaches to multiple testing. *European Journal of Epidemiology* 2019; 34:809-21

Reviewer 2

Dr Lara Lemon, University of Pittsburgh

Authors used a clever approach to compare the change in NICU utilization in the quarter pre- and post-COVID to the previous seven-year changes. It is an important paper with strong methods.

Author response: Thank you

2.1 Limit the number of stratifications. Table 1 is hard to follow and understand the impact with the variety of stratifications. I suggest limiting the gestational age stratifications to: Term, PTB (<37), and VPTB. Using the GA1, etc abbreviations is also confusing to the reader. Or limit the IMD stratifications. When only the significant changes are displayed it is hard to understand overall effects

also. For instance: What happened to breast feeding overall? I would consider recreating Table 1 with all changes assessed, not only those 'unusual', after only a select few stratifications are chosen. Justification for which are chosen should be provided.

Author response: We would prefer not to limit the gestational age stratifications to term, PTB (<37), and VPTB, but to retain the World Health Organisation categories (extremely preterm GA1: <28⁺⁰; very preterm GA2: 28⁺⁰ to 31⁺⁶; moderate to late preterm GA3: 32⁺⁰ to 36⁺⁶; and full term GA4: ≥37⁺⁰ weeks^{+days}). We have explained why in the first paragraph of the "Discussion" (*We assessed all preterm and full-term admissions as well as extremely preterm, very preterm, and moderate-to-late preterm groups individually, as degree of immaturity has a cardinal influence upon care pathways and morbidities*). We have however replaced the abbreviations GA1, GA2, GA3, GA4 with the terms extremely preterm, very preterm, moderate to late preterm and full term respectively, throughout the paper, which we hope will make for easier reading.

We would also prefer not to limit the IMD stratifications as to reduce them beyond five, would reduce the sensitivity of any inferences.

We have provided justification for the outcomes selected (changes that were both highly unusual and met our strength of evidence threshold). Table 1 would be very large, and we believe, confusing, were we to include all outcomes; however we have provided all the additional outcomes not shown in Table 1 in a new Supplementary Table.

2.2 Did you run a sensitivity analysis excluding the covid+ deliveries?

Author response: We did not as there were very few confirmed COVID-positive deliveries (please see Table 2), hence excluding them would not make any difference.

2.3 I believe your findings of increases in NICU admissions for Black infants is novel and should be highlighted more. Again, (re: MC1) was there an increase in Black babies' admissions regardless of term? I wonder because black women have been seen to have the least impact regarding preterm decreases.

Author response: Fig 3 shows that there were increases from winter to spring in admissions of Black infants in extremely preterm, moderate-to-late preterm, and full-term categories, but not the very preterm category. The summary changes were an increase in all preterm categories, and all admissions combined. Of these increases, only that for the full-term category, and all admissions combined reached our threshold for significance.

2.4 Please describe if you were able to account for spontaneous vs indicated deliveries. If so, this is a very important aspect that should be used in your analysis. Though the 'cause of preterm birth' on page 4 line 40, is largely unknown, discerning between spontaneous and indicated is possible in many data sets. Were authors able to determine this difference?

Author response: We were unable to distinguish between spontaneous and medically indicated preterm deliveries. We have noted this limitation in the "Discussion" (*"...though we were unable to distinguish between spontaneous and medically indicated preterm births"*).

2.5 Please explain a bit further if this would include infants admitted directly from delivery? I believe so but am not certain.

Author response: Yes, admissions to NHS neonatal units are predominantly directly from the delivery suite of the same hospital, but also include postnatal transfers from other neonatal units.

2.6 Authors have opposite finding from previous studies in US- demonstrating less preterm in the most deprived groups. Is it possible mandates in UK had differential impact compared to those in the US? A sentence on this in the discussion on this is warranted.

Author response: We have included the following sentence in the "Discussion": *"Comparisons between the UK and US are problematic, first because the healthcare systems are very different, and second, because US reports are centre rather than population-based, and hence at risk of ascertainment bias"*.

2.7 Page 3, final bullet point: No gestational age? Or no delivery data?

Author response: No data were available on births by gestational age.

2.8 Would a period effect say from 2013-2015 then have a big impact with this approach to estimate the background?

Author response: This would be plausible if there had been an event comparable in effect to Covid in the period 2012-2019. However, our analysis is predicated on the absence of any event of such magnitude during this period, so that background influences that cause fluctuations in outcome rates are set against an influence that is by general consensus highly exceptional.

2.9 Methods on page 6 lines 14-30 are confusing. Specifically, how are there 14 pre-COVID differences? I understand 14 time points being both quarters (2013-2020), but only 7 differences, no? Or was each compared to post-COVID? Please elaborate.

Author response: Thank you for pointing out this confusion. We have revised the text to be more specific as follows: “For each measure and gestational age category, we held out the two three-month periods for the COVID difference (i.e. the spring (Apr 2020-Jun 2020) and winter (Dec 2019-Feb 2020) periods). We then used the 14 corresponding pre-COVID spring and winter three-month periods to estimate the seven background spring-winter differences against which to assess the COVID spring-winter difference. For the 14 pre-COVID three-month periods we identified posterior distributions over the binomial probabilities, approximating them with Gaussian distributions by moment matching and applying shrinkage assuming the individual three-month rates are drawn from a common distribution”.

2.10 To support page 8, lines 8-13: do you have the diagnoses and chief complaint for the admission?

Author response: We regret these details are not available in the database.

2.11 Page 9 lines 57-page 10 lines 5: Please make more clear. I cannot understand this sentence. Are authors claiming that late care caused an increase in term NICU admissions though overall lower clinical thresholds for admission were a good thing?

Author response: We suggest (we do not claim) that delayed presentation to a healthcare facility might increase the likelihood of a problematic delivery, and would be a plausible explanation for the highly unusual increase in transfers of moderate-to-late preterm and full-term babies to a higher designation neonatal unit. Additionally, anxieties about COVID might also have altered thresholds for admission of more mature babies (for whom such decisions often have a degree of subjectivity in contrast to the universal admission to neonatal care of extremely preterm babies), which would be a plausible explanation for the overall highly unusual fall in full-term admissions during the immediate COVID-19 period.

2.12 SC9. Do you have any data on the ed use? This would be an interesting addition.

Author response: Our apologies; what is “ed use”? Is the reviewer referring to “Emergency Department” use? If so, no, these data were not available.

2.13 Did previous period changes include or exclude March?

Author response: We excluded March as stated in the first paragraph of the “Analyses” section.

2.14 I suggest changing ‘spring-winter’ to ‘winter-spring’ throughout as this is the chronological order for the comparison. Same with table 1. The change is negative because we are going winter to spring, correct?

Author response: The change is calculated as the difference “Spring minus winter” (hence negative as admissions were lower in the spring period). We have added the text “(i.e. *spring minus winter difference*)” to the “Analyses” section to make this clear.

VERSION 2 – REVIEW

REVIEWER	Lara Lemon University of Pittsburgh, Obstetrics, Gynecology and Reproductive Medicine
REVIEW RETURNED	16-Sep-2021

GENERAL COMMENTS	Wonderful manuscript. Well thought out methods and clearly described. My only minor comment is to include a description of IMD. The results on Page 8, Line 9, need to remind the reader if IMD quintile 1 is advantaged or disadvantaged.
--